# Revealing Long-Term Indoor Air Quality Prediction: An Intelligent Informer-Based Approach

**DOI:** 10.3390/s23188003

**Published:** 2023-09-21

**Authors:** Hui Long, Jueling Luo, Yalu Zhang, Shijie Li, Si Xie, Haodong Ma, Haonan Zhang

**Affiliations:** 1College of Information Science and Engineering, Changsha Normal University, Changsha 410199, China; luojueling@gmail.com (J.L.); zhangyalu1211@gmail.com (Y.Z.); lishijie925@gmail.com (S.L.); xiesi102741@gmail.com (S.X.); mahaodong0328@gmail.com (H.M.); zhanghaonan52168@gmail.com (H.Z.); 2Broad Air-Conditioning Co., Ltd., Postdoctoral Workstation, Changsha 410138, China

**Keywords:** artificial intelligence, indoor air quality, public health, indoor environment quality, public health

## Abstract

Indoor air pollution is an urgent issue, posing a significant threat to the health of indoor workers and residents. Individuals engaged in indoor occupations typically spend an average of around 21 h per day in enclosed spaces, while residents spend approximately 13 h indoors on average. Accurately predicting indoor air quality is crucial for the well-being of indoor workers and frequent home dwellers. Despite the development of numerous methods for indoor air quality prediction, the task remains challenging, especially under constraints of limited air quality data collection points. To address this issue, we propose a neural network capable of capturing time dependencies and correlations among data indicators, which integrates the informer model with a data-correlation feature extractor based on MLP. In the experiments of this study, we employ the Informer model to predict indoor air quality in an industrial park in Changsha, Hunan Province, China. The model utilizes indoor and outdoor temperature, humidity, and outdoor particulate matter (PM) values to forecast future indoor particle levels. Experimental results demonstrate the superiority of the Informer model over other methods for both long-term and short-term indoor air quality predictions. The model we propose holds significant implications for safeguarding personal health and well-being, as well as advancing indoor air quality management practices.

## 1. Introduction

Clean air is not just a luxury but a fundamental necessity for human survival. Even a brief interruption in breathing can have severe consequences, leading to biological death. Astonishingly, research conducted by esteemed institutions such as the U.S. Environmental Protection Agency and Health Canada has unveiled a startling fact—approximately 68% of human diseases are linked to air pollution [1]. Despite its critical importance, indoor air quality often receives inadequate attention. People spend an average of 21 h indoors daily, making the air they breathe a crucial factor in their health and safety. It is not just the outdoor environment that needs clean air, indoor air quality is equally essential, especially considering the amount of time individuals spend inside buildings and homes. PM2.5 is harmful to the human respiratory system, cardiovascular system, and immune system. In severe cases, it may lead to an increased risk of cancer [2]. This eye-opening statistic underscores the urgent need to reduce air pollution globally, as it has the potential to significantly decrease hospitalizations and enhance overall well-being.

In 2019, a groundbreaking study was conducted by the Chemical Institute of China National Testing & Inspection Institute in Sichuan Province. The study analyzed data from 6482 indoor locations and published the “2019 White Paper on Indoor Air Pollution in China”. The findings were concerning, revealing that a staggering 95% of indoor spaces failed to meet air quality standards within the first year of renovation [3]. Furthermore, the non-compliance rate remained high at 55% even after three years. The situation was particularly alarming in offices, where individuals spend a substantial portion of their time, with a non-compliance rate reaching as high as 90% [4].

## 2. Related Work

Air quality is a critical matter of concern in terms of the impact on public health and well-being. Numerous research studies state the adverse health effects associated with poor air quality levels, which include premature death, and respiratory and cardiovascular disease, along with a relevant increase in asthma attacks, dementia, and cancer [5]. According to the EPA, indoor levels of pollutants may be up to 100 times higher than outdoor pollutant levels and have been ranked among the top 5 environmental risks to the public [6]. Thus, accurate prediction of air pollutant concentrations is crucial for management authorities and vulnerable populations to minimize the exposure to hazardous pollutants.

In the past few years, several methods have been proposed to predict indoor air quality. In general, indoor air prediction methods can be divided into two categories: traditional machine learning methods and deep learning methods. 

For its powerful and fast predictions with big data, machine learning has gained tremendous popularity. Some researchers have applied machine learning algorithms to the short- and long-term prediction of air quality successfully. For example, Boznar M et al. developed a method based on the neural network for prediction for SO_2_ pollution around the biggest Slovenian thermal power plant at Sostanj [7]. Nieto P J G et al. built a regression model of air quality by using the support vector machine (SVM) technique in the Avilés urban area (Spain) at local scale [8]. Liu H et al. used a data-driven prediction method based on a mixed GPR to predict the PM2.5 concentration in a subway station [9]. Taheri S et al. used machine learning algorithms to forecast CO_2_ concentrations across a range of forecasting horizons [10].

However, their performances are generally limited in practical application. This is because the shallow structures of traditional machine learning models often make them fail to capture the complex and nonstationary fluctuations of air pollutant data, resulting in their poor performance in long-term air pollutant prediction, which is crucial to preemptive air pollutant prevention and management.

With the rapid growth of artificial intelligence, deep learning (DL) technologies, which have deeper structures and more powerful learning abilities, have been extremely used in the field of air pollution prediction [11]. A large number of researchers have applied deep learning to outdoor air quality prediction, such as linear regression [12], recurrent neural networks (RNN) [13], LSTM [14], and GRU [15], among others. They can not only predict specific values of pollutant concentrations, but also classify pollutant levels as well as achieve long-term prediction tasks for multiple days in the future. For example, Monteil J et al., integrated deep learning together with techniques from PDE-based domain decomposition to demonstrate a framework for air-pollution monitoring and forecasting at Dublin, Ireland [16]. 

In the realm of indoor air quality prediction, Allen et al. conducted a study where they monitored indoor particle matter and employed regression models to predict air quality [17]. Zhao expanded on this approach by integrating external seasonal conditions into regression models, incorporating a recurrent neural network (RNN) with long short-term memory units as the framework. Their work sought to improve the accuracy of air quality prediction by considering additional environmental factors [18].

Athira V et al. used seq2seq with LSTM and GRU for air quality prediction and demonstrated that seq2seq is a promising method for air quality prediction [19]. For heating, ventilation, and air conditioning (HVAC) systems affected by multiple environmental variables, a deep learning model based on the main architectures of LSTM and GRU was proposed to predict future microclimate conditions with optimization of energy consumption, achieving advanced multivariate time series prediction of indoor air quality [20].

Ahn jae-hyun et al. proposed a complex single-step multivariate time series indoor air quality prediction model. The model aims to exploit the complex relationships between different air quality parameters and improve the prediction ability [21].

Yeo-Kyung Lee’s research aims to improve the indoor air quality of office spaces, especially considering the air quality of indoor office people for a long time and the prevention of airborne diseases [22]. They developed an “Office Space CO_2_ Prediction and IAQ Improvement Suggestion Tool” to help non-experts manage IAQ and reduce the harm of air quality to indoor office workers.

Ali Majdi et al. published a study focusing on air quality control in smart buildings. In this study, a neural network with radial basis functions was used to predict the indoor air quality of a commercial office center in Mashhad, Iran [23].

Liang Y. and colleagues proposed an innovative model called AirFormer for predicting air quality across thousands of locations in mainland China with unprecedented fine spatial granularity. The model utilizes 4 years of data from 1085 monitoring stations in China. Compared to previous models, AirFormer reduces the prediction error and improves the forecast for the next 72 h by 5% to 8% [24].

However, on the other hand, the ability to capture the time-dependency and correlation between data indicators in these methods still needs improvement. In practical applications, they only focus on the temporal information within air quality data, while overlooking the interdependencies among monitored data indicators.

In summary, research on PM2.5 prediction in indoor and outdoor environments as well as long sequence prediction of indoor air quality is limited [25], but there is a high demand for long-term sequence prediction in practical applications [26]. This holds great significance for indoor air quality monitoring and management [27,28,29]. If more accurate prediction results can be provided, it will help people take timely action to improve indoor environments, thereby enhancing the quality of life and health [5,10].

Therefore, in this study, the informer model was employed for conducting long sequence prediction of indoor air quality, aiming to validate the informer’s performance in predicting long sequences of indoor air quality. Additionally, specific network structures were integrated to enhance the model’s capability to learn correlations between different data indicators. The main contributions of this research are summarized as follows:We proposed an informer-based indoor air quality prediction model that not only leverages informer’s advantage in capturing long sequence time dependencies but also has the ability to capture correlations between air quality data indicators, thereby enhancing the accuracy of indoor air quality prediction;Through experiments on real datasets, our model demonstrated remarkable performance in predicting indoor PM2.5 concentrations across different prediction timeframes;The experiments indicate that our model can accurately predict not only PM2.5 concentrations, but also effectively forecast other air quality data. Furthermore, our model exhibits adaptability to prediction tasks with fewer data dimensions, such as long sequence prediction tasks that do not consider significant spatial dependencies and involve high data correlation.

The remainder of this paper is organized as follows: In Section 3, we provide a detailed description of the model architecture and its individual components. In Section 4, we present the dataset, experimental setup, and experimental results. Finally, in Section 5, we present our conclusions.

## 3. Methodology

### 3.1. Overview of Informer

The overall architecture of Informer is illustrated in Figure 1. Informer primarily consists of two components: (1) MLP, which extracts valuable feature information from the raw monitoring data (i.e., indoor and outdoor air quality data); (2) the Informer layer, which captures the long sequential temporal dependencies within the historical data.

### 3.2. MLP Layer

Taking into consideration the influence of outdoor air quality on indoor air quality, as well as the interrelation among various data indicators, this study employs an MLP to extract features from the raw air quality and meteorological data as input for the Informer layer. For instance, taking indoor PM2.5 as the predictive target, the historical indoor and outdoor PM2.5, as well as temperature and humidity data, are represented as $X_{n}\in R^{1\times 1\times L}$, while the corresponding feature data of dimension m (excluding indoor PM2.5) is represented as $F_{n}\in R^{1\times M\times L}$. To acquire enhanced feature representations, we initially input $F_{n}\in R^{1\times M\times L}$ into the MLP, extracting improved feature representations $F_{e}\in R^{N\times 1\times L}$ from the m-dimensional feature data.

### 3.3. Informer Layer

In the indoor air quality prediction task, capturing the temporal dependencies within the data plays a crucial role. This paper introduces the Informer model based on the Transformer architecture to capture these temporal dependencies. Informer adopts a similar encoder-decoder architecture; however, it enhances the original Transformer framework’s performance on long sequence data prediction and reduces time complexity and memory usage by incorporating the ProbSparse self-attention mechanism and self-attention extraction operations. Additionally, Informer employs a dynamic inference method for predictive output.

The Informer’s encoder takes long sequence data as input and captures the temporal dependencies within the data using the ProbSparse self-attention mechanism and self-attention extraction operations. Probability-based self-attention: In comparison to the original self-attention mechanism in Transformers, the ProbSparse self-attention mechanism effectively reduces the time complexity. In the self-attention mechanism, each element is paired with all other elements to determine the attention distribution, introducing heavy computational burden and memory consumption when dealing with long sequence data. However, in most real-world scenarios, the attention distribution is dominated by only a few important query-key pairs.

To measure the similarity of attention distributions, Informer utilized Kullback-Leibler (KL) divergence as a metric [30]. KL divergence can be used to compare differences between probability distributions. In this improved self-attention mechanism, the sparsity measure of the *i*-th query is defined as follows:(1)$$M(q_{i},K)=ln\sum_{j=1}^{LK}exp\left(\frac{q_{i}k_{j}^{T}}{\sqrt{d}}\right)-\frac{1}{LK}\sum_{j=1}^{LK}\frac{q_{i}k_{j}^{T}}{\sqrt{d}}$$
where the $M\left(q_{i},K\right)$ consists of two components: the LogSumExp (LSE) of q_iacross all keys, and the arithmetic mean of the resulting values. A larger value of $M\left(q_{i},K\right)$ for the *i*-th query indicates a more ‘diverse’ attention probability p. This increased diversity leads to a higher likelihood of including the dominant dot-product pairs in the header field of the long-tail self-attention distribution [31].

If the sparsity measure $M\left(q_{i},K\right)$ of the *i*-th query is large, its attention probability p is more “diverse” and likely to include the important dot products in the head of the long-tail attention distribution.

In order to identify the important $q_{i}$, Informer’s ProbAttention utilizes a threshold to determine whether each query exceeds it, thereby assessing its significance. Queries that surpass the threshold are considered important and selected as the target of key attention.
(2)$$\overset{-}{M}(q_{i},K)=\underset{j}{\max}\left\{\frac{q_{i}k_{j}^{T}}{\sqrt{d}}\right\}-\frac{1}{LK}\sum_{j=1}^{LK}\frac{q_{i}k_{j}^{T}}{\sqrt{d}}$$
where $\underset{j}{max}$ is less affected by zero values and maintains numerical stability. In practice, the input lengths of queries and keys are usually the same in self-attention computation, denoted as $L_{Q}=L_{K}=L$, resulting in a total ProbSparse self-attention time and space complexity of $O\left(LlnL\right)$ [31].

Based on the informer proposed measurement [31], the ProbSparse self-attention is achieved by restricting each key to attend only to the u most dominant queries:(3)$$A\left(Q,K,V\right)=Softmax\left(\frac{\overset{-}{Q}K^{T}}{\sqrt{d}}\right)V$$
where $\overset{-}{Q}$ is a sparse matrix of the same size as q, which comprises only the Top-u queries determined by the sparsity measurement $M\left(q,K\right)$.

By incorporating this threshold-based approach, Informer effectively identifies and prioritizes the queries with higher importance, enabling the model to focus on key attention. This selection process helps in capturing the most relevant information for improved prediction and enhances the overall performance of the Informer model.

The encoder is the most crucial component of Informer as it extracts important temporal dependencies from the input time series and computes feature attention weights from different perspectives using multi-head attention. In the encoder, the input sequence $X_{t}$ undergoes processing and is transformed into a matrix $X_{t}^{e}n\in R^{\left(L_{x}\times d_{model}\right)}$, where $L_{x}$ represents the length of the input sequence and $d_{model}$ is the dimension of the model.

Another crucial aspect of the Informer encoder is self-attention distilling. It enhances the quality and focus of the encoder’s feature maps, further improving computational efficiency.
(4)$$X_{t}^{\left(j+1\right)}=MaxPool\left(ELU\left(Conv1d\left({\left[X_{t}^{j}\right]}_{A}B\right)\right)\right)$$
where ${\left[\cdot \right]}_{AB}$ represents an attention block that includes multi-head ProbSparse self-attention and other necessary operations. *Conv1d* denotes a 1D convolution operation along the time dimension, utilizing $ELU\left(\cdot \right)$ as the activation function. MaxPool represents a max pooling operation with a stride of 2, used for downsampling and halving the output.

By incorporating this threshold-based approach, Informer effectively identifies and prioritizes the queries with higher importance, enabling the model to focus on key attention. This selection process helps in capturing the most relevant information for improved prediction and enhances the overall performance of the Informer model.

The Decoder of the Informer model is capable of generating long sequence outputs in a single forward pass. It adopts a standard decoder structure consisting of a stack of two identical multi-head attention layers. However, the Decoder of Informer differs from traditional decoder architectures as it employs a generative inference approach to address the speed degradation issue in long sequence prediction.

In the Decoder, the input vector is represented as $Xt_{de}$, which is formed by concatenating the start token $Xt_{token}$ and a placeholder $Xt_{0}$ for the target sequence. The start token $Xt_{token}$ is a vector of length $L_{token}$, representing the initial input of the decoder. The placeholder $Xt_{0}$ for the target sequence is a vector of size $Ly\times d_{model}$, with each scalar value set to 0.

In ProbSparse self-attention calculations, a masked multi-head attention mechanism is utilized, where the masked dot products are set to negative infinity (−∞) to prevent each position from attending to future positions, thus avoiding the autoregressive issue. The final output is obtained through a fully connected layer, with the output dimension dy being multi-dimensional as defined in the paper.

Generative inference is an important technique in Informer, inspired by the “dynamic decoding” method in the field of natural language processing, and extended to a generative approach. In generative inference, instead of selecting a specific token as the start token, a randomly sampled subsequence of length $L_{token}$ from the input sequence is used as the start token. The generative inference decoder of Informer also includes the timestamps $X_{0}$ of the target sequence, which provide contextual information about the target week. This allows the decoder to generate long sequence outputs in a single forward pass without the need for time-consuming “dynamic decoding” as in traditional decoders.

### 3.4. Evaluation Metrics

In our research, we utilized two key loss functions—Mean Absolute Error (MAE) and Mean Squared Error (MSE)—to train and assess our indoor air quality prediction model. MAE measures the average absolute difference between predicted and actual values, providing robustness against outliers and unexpected fluctuations in air quality. On the other hand, MSE, calculating the average squared difference, offers a more sensitive evaluation of larger errors. The combination of MAE and MSE allowed us to holistically evaluate our model’s predictive performance, considering both absolute and squared errors. This approach enabled us to strike a balance between minimizing overall prediction errors and handling extreme cases effectively, thus ensuring strong generalization capabilities across diverse indoor air quality conditions, which are expressed as: (5)$$MSE=\frac{1}{N}\sum_{i=1}^{N}{\left(y_{pred_{i}}-y_{true_{i}}\right)}^{2}$$
(6)$$MAE=\frac{1}{N}\sum {\left(i=1\right)}^{N}\left|y_{pred_{i}}-y_{true_{i}}\right|$$
where $y_{pred_{i}}$ and $y_{true_{i}}$ denote the predicted value and ground truth of an indoor air quality (e.g., PM2.5). The MSE and MAE are used to measure prediction errors. A smaller value of MSE or MAE indicates better predictive performance.

## 4. Experiment

### 4.1. Dataset and Hyperparameter

#### 4.1.1. Dataset

The dataset originates from the Quality Control Building of Yuanda Central Air Conditioning Park in Changsha County. It encompasses air quality data for six parameters, including indoor and outdoor temperature and humidity, as well as indoor and outdoor PM2.5 concentrations, from January to December of 2019. The sampling frequency is once every hour.

The purpose of collecting this dataset is to investigate and monitor the relationship between fine particulate matter (PM2.5) concentration levels in the air and temperature and humidity. These data allow us to comprehend and analyze the variations and distribution patterns of particles within this size range. The distribution of indoor PM levels in this dataset exhibits a non-uniform pattern, characterized by the presence of numerous outliers. This unique distribution poses challenges when attempting long sequence forecasting.

After collecting the air quality data, preprocessing is necessary. On one hand, due to issues such as sensor malfunctions and equipment maintenance interruptions at monitoring points, the air quality data for each monitoring point contain missing values. In this study, a linear interpolation method is employed to fill in the missing values in the air quality data for each monitoring point a common and straightforward strategy used in related works [32]. As a result, each monitoring point has 14,005 indoor and outdoor air quality records, including PM_2.5_, CO_2_, Temperature, and Relative Humidity.

On the other hand, to mitigate the impact of different dimensions of data on model training and accelerate the convergence speed of the model, standardization is applied to various features of both air quality and meteorological data. This standardization can be expressed as: (7)$$z=\frac{x-\mu }{\sigma }$$
where *x* and *z* represent the original data and standardized data, respectively, *μ* is the mean of *x*, and *σ* is the standard deviation of *x*. As a result, the distribution of each feature is scaled to have a mean of 0 and a standard deviation of 1.

#### 4.1.2. Experimental Parameter Configuration

In this paper, the Informer model is optimized using the Adam optimizer, with the learning rate initialized at 1 × 10^−4^ and halved at the end of each epoch. The training process consists of 8 epochs, and an early stopping strategy is implemented to improve efficiency. The model’s encoder is constructed with 4 layers, while the decoder comprises 2 layers. The batch size is set to 32 during the training process.

Our Informer model consists primarily of two parts: MLP layers and Informer layers. The implementation details of each part are described below.

Firstly, the MLP layers consist of four fully connected layers, with an input feature data dimension of 6 and a hidden layer of 100 neurons, the latter being the number of neurons in the final layer. The Informer layers consist of 4 encoder and 2 decoder layers, with a hidden dimension of 512. In both the encoder and decoder layers, the ProbSparse self-attention mechanism is implemented, along with residual connections, feedforward networks with 2048 inner neurons, and a Dropout layer with a dropout rate of 0.01. The attention head count is set to 8.

The learning rate is set to 1 × 10^−4^, batch size is 32, and the model is trained using mean squared error loss and the Adam optimizer. The monitoring point data consists of air quality data for 12 months; hence, the training/validation/testing time split is 8/2/2 months. The total number of epochs is set to 100, and an early stopping strategy with a patience of 5 is employed to train the model. For a more detailed list of parameters, please refer to Table 1.

All experiments were conducted on the windows server with an Intel i9 9900k CPU (5.0 GHz, 16 M RAM) and an Nvidia RTX 2080ti GPU (11G GPU RAM).

### 4.2. Experimental Results

#### 4.2.1. Performance Comparison

To demonstrate the effectiveness of the proposed model, we compared it with three different baseline methods. All methods were trained and tested using the same training, validation, and testing datasets. The details of the baselines are as follows:

DNN (Deep Neural Network): DNN is an artificial neural network composed of multiple fully connected layers with sigmoid activation functions. It is used to uncover relationships between historical and future data.

LSTM (Long Short-Term Memory) [33]: LSTM is a type of recurrent neural network (RNN) designed to address the vanishing gradient problem in traditional RNNs. It is capable of learning long-term dependencies in sequential data by selectively remembering or forgetting information.

GRU (Gated Recurrent Unit) [34]: GRU is similar to an LSTM with a forget gate, but it has fewer parameters since it lacks an output gate. GRU performs similarly to LSTM in tasks such as polyphonic music modeling, speech signal modeling, and natural language processing.

Transformer [35]: the Transformer is constructed with an encoder and a decoder, both equipped with self-attention mechanisms, enabling it to capture long-range dependencies within sequence data.

To ensure that the prediction results are not biased towards specific randomly initialized weight parameters, each method was run three times, and Table 2 reports the average metrics for each method. Table 2 presents the performance of the Informer model and other baseline models in predicting PM2.5 air pollutant concentrations for 6 h, 12 h, and 24 h ahead. We can observe that our Informer model significantly outperforms other baselines in terms of MSE and MAE. Furthermore, performance of deep learning methods based on attention mechanisms generally surpasses that of traditional deep learning models for time series data. This suggests that attention-based methods have better capabilities in handling complex and nonlinear time series air quality data.

#### 4.2.2. Model Fitting Results

By fitting the model with the hyperparameters from Table 1, the Informer model demonstrated excellent predictive performance on the test dataset, as shown in Figure 2. The results clearly indicate that the model’s predictions closely align with the actual values of indoor PM (Particulate Matter) concentrations.

The Informer architecture’s ability to capture and leverage temporal dependencies in the indoor air quality data plays a crucial role in achieving accurate predictions. Its self-attention mechanism and feature extraction capabilities allow it to effectively capture complex patterns and relationships within the data, enabling it to make precise forecasts.

Moreover, the model’s performance on the test dataset underscores its robustness and generalization ability. It successfully handles the challenges posed by the uneven distribution and outliers in the data, providing reliable predictions even in challenging scenarios.

## 5. Conclusions

In this study, we introduce an indoor air quality prediction model that excels in indoor air quality forecasting. To harness the temporal correlations and data dependencies in air quality data, the model employs a feature extraction layer based on MLP to learn the data correlations among various monitoring indicators. This extraction layer is designed to capture both short-term and long-term temporal information and data correlations within air quality data. Moreover, the main objective of the MLP-based feature extraction layer is to learn data correlations from indoor air quality data nodes and outdoor air quality data nodes. In essence, this design captures the data dependency and temporal dependency issues between indoor air quality and closely situated outdoor nodes, leading to more accurate predictions.

Experimental results demonstrate that our proposed air quality prediction model outperforms other methods in predicting both short-term and long-term air quality data. The indoor air quality model presented in this paper holds significant implications for safeguarding personal health and well-being, as well as advancing indoor air quality management practices. 

## Figures and Tables

**Figure 1 sensors-23-08003-f001:**
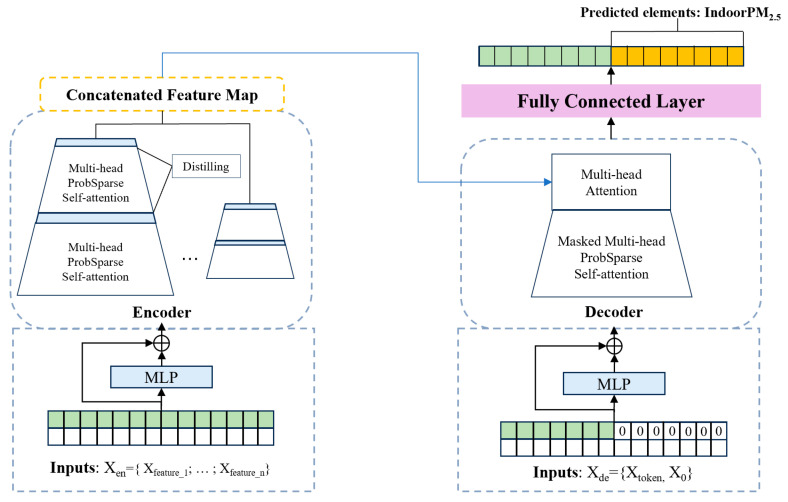
Diagram of the Informer Architecture.

**Figure 2 sensors-23-08003-f002:**
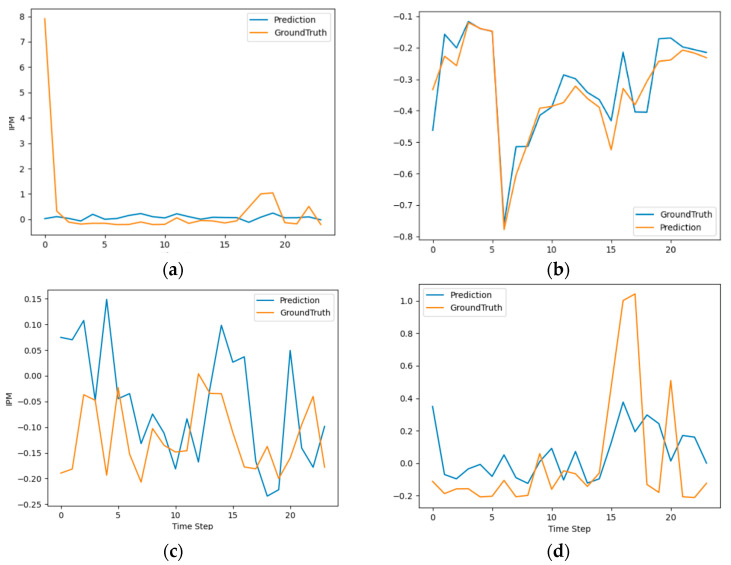
The predicted variations of indoor PM values exhibit significant fluctuations, with numerous outliers. Overall, the model closely approximates the true distribution trends in the following ways: (**a**) The model demonstrates a relatively close fit to the data but fails to predict sudden spikes in certain trends; (**b**) The model nearly perfectly predicts the data; (**c**) The model accurately forecasts the overall trends, but lags in predicting specific time steps; (**d**) The model correctly predicts the trends but encounters significant fluctuations, struggling to handle extreme outlier points.

**Table 1 sensors-23-08003-t001:** Hyperparameters.

Hyper-Parameter	Value
Learn_rate	1 × 10^−4^
optimizer	Adam
Batch	32
attn	8
patience	3
Epoch	50
seq_len	96
label_len	48
Pred_len	24
E_layers	4
D_layers	2

**Table 2 sensors-23-08003-t002:** Comparison of informer and other baselines in predicting PM2.5 atmospheric pollutant concentrations.

	24 h		12 h		6 h	
	MSE	MAE	MSE	MAE	MSE	MAE
DNN	0.492	0.531	0.399	0.487	0.396	0.502
LSTM	0.203	0.398	0.204	0.393	0.317	0.443
GRU	0.143	0.301	0.197	0.391	0.213	0.387
Transformer	0.097	0.224	0.167	0.313	0.209	0.381
Informer (Our)	0.017	0.023	0.160	0.157	0.227	0.392

## Data Availability

3rd Party Data: Restrictions apply to the availability of these data. Data was obtained from Broad Air-Conditioning Co., Ltd and are available the author with the permission of Broad Air-Conditioning Co., Ltd.

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
