# Peer review of "Revealing Long-Term Indoor Air Quality Prediction: An Intelligent Informer-Based Approach"

_sensors, 2023, doi:10.3390/s23188003_

Round 1
Reviewer 1 Report
The paper presents a novel approach to predict long-term indoor air quality using the Informer model. The presented idea seems to be great, and this is confirmed by the reported results. The prediction quality measured by MSE and MAE is better than for other methods. The limitations and typical weaknesses of prediction were listed soundly. However, the method is described too scantly and the reader must guess some details.
Abstract. Please, explain acronym PM or do not use such acronyms in the abstract.
Line 97. There is no verb in the sentence!
Line 100. It should be a space before the bracket “health[18, 20]”.
Equations 1, 2 and 3. Please explain all the symbols.
Section 4.1.2. Hyperparameters. Did you tune these parameters in any way? Please add some words on their impact on prediction quality.
Section 4.2.1. ProbAttention effect. What does ‘Q’ mean?
Figure 2 is hardly readable. Please explain its meaning.
Line 272 and further. The description is somewhat unclear: does or not shorter prediction lengths affect MSE and MAE?
Author Response
Subject: Gratitude for Reviewer Feedback on My Manuscript
Dear Reviewer's,
I hope this message finds you well. I wanted to express my heartfelt gratitude for the valuable feedback you provided on my manuscript titled "Revealing Long-Term Indoor Air Quality Prediction: An Intelligent Informer-Based Approach" Your insightful comments and suggestions have been incredibly helpful in shaping the direction and content of my work.
Your thoughtful and constructive feedback has provided me with a fresh perspective and has greatly contributed to the improvement of my manuscript. Your input has guided me to make necessary revisions that have undoubtedly enhanced the overall quality of the article.
I truly appreciate the time and effort you dedicated to reviewing my work. Your expertise and attention to detail have played a pivotal role in refining the ideas and arguments presented in the manuscript.
Once again, thank you for your dedication to the peer-review process and for your invaluable input. Your feedback has been an instrumental part of this publication journey.
Sincerely,
Jueling Luo

Reviewer 2 Report
Please see the attachment.

Moderate editing of English language required
Author Response

(The authors gave the same response as above.)

Round 2
Reviewer 1 Report
Since the Authors have addressed all my comments I have no further remarks
Author Response
Dear Reviewer,
I hope this message finds you well. I am writing to extend my deepest gratitude and respect for your invaluable contribution as the reviewer of my paper.
Your insightful comments and suggestions have played a pivotal role in advancing my research. Through your review, I was able to identify and address many aspects that I hadn't previously considered, resulting in a significant improvement in the content, structure, and presentation of my paper.
I am fully aware of the time and effort you dedicated to the review process, and I am truly impressed by your professionalism. Your expertise and patient guidance have been immensely beneficial to me, and I will carry the lessons learned throughout my academic journey.
Warm regards,
Jueling Luo
Reviewer 2 Report
Thanks for the corrections.
There are still some minor items which should be corrected,
1. The references in the text are not in order, the first reference is [23]. Please correct this subject throughout the text.
2. The formulas should be typed in italics.
3. The quality of figures should be improved.
Minor editing of English language required
Author Response
Response to reviewers
Dear Reviewer,
I would like to express my sincere gratitude for the valuable feedback and suggestions you provided during the review of my manuscript. Your professional insights have greatly enhanced the quality of my work and I have gained invaluable insights for my academic research.
Thank you for your patient review and insightful comments. Under your guidance, I now have a clearer understanding of my research and feel more confident in presenting it to readers.
I greatly appreciate your time and dedicated effort. Your assistance is immeasurable to me. I have made the necessary revisions to my manuscript based on your suggestions, and I hope these changes meet your expectations.
Once again, thank you for your expert feedback.
Warm regards,
Jueling Luo
Comment 1: The references in the text are not in order, the first reference is [23]. Please correct this subject throughout the text.
Response: Thanks for your suggestion, we have corrected it.
Comment 2: The formulas should be typed in italics.
Response: We appreciate your suggestion; we have corrected it.
Comment 3: The quality of figures should be improved.
Response: Thank you very much for your suggestion. We have made our best efforts to enhance the quality of the figures as per your recommendation. Your feedback is greatly appreciated.